# Assessing the Impact of COVID-19 on Consumer Food Safety Perceptions—A Choice-Based Willingness to Pay Study

**Oliver Meixner \*** and **Felix Katt**

Institute of Marketing and Innovation, Department of Economics and Social Sciences, University of Natural Resources and Life Sciences, A-1180 Vienna, Austria; felix.m.katt@gmail.com
\* Correspondence: oliver.meixner@boku.ac.at; Tel.: +43-1-47654-73515; Fax: +43-1-47654-73509

**Abstract:** As the COVID-19 pandemic brings about sudden change in societies across the globe and likely heralds the start of a recession, we examine the pandemic's impact on consumer food safety perceptions. Due to its origin, COVID-19, likely spurring from an animal-to-human transmission in the context of a wet market, may impact consumer food perceptions in similar ways to the avian flu (H5N1) and the swine flu (H1N1). We examine this effect by studying preferences for beef meat in a consumer survey in the United States ($n = 999$) using a choice-based experiment. We compare our findings to Lim et al. (2014), who elicited consumer beef willingness to pay (WTP). Additionally, we investigate the impact of the looming recession by analyzing several attributes and their effect on consumer preferences. Our findings suggest that food safety concerns have become more important. As a result, production standards and the country of origin have lost importance. Additionally, we show that the socioeconomic impact for some respondents impacts their shopping preferences. Finally, we outline potential areas for future research as well as managerial implications.

**Keywords:** choice-based conjoint analysis (CBCA); willingness to pay (WTP); food safety; COVID-19; coronavirus

## 1. Introduction

When the World Health Organization declared the novel coronavirus (COVID-19) a pandemic, governments across the globe took restrictive action that is likely to lead to a severe economic recession resulting in increased unemployment and decreased economic welfare [1]. COVID-19, likely originating from an animal-to-human transmission in the context of a wet market [2], will undoubtedly have widespread effects on society and consumers. It is therefore our aim to examine and understand the impact COVID-19 may have on consumer perceptions of food safety in general and for meat in particular. To achieve this aim we seek to replicate parts of the study by Lim et al. [3], who examined US consumers' willingness to pay (WTP) for beef with an emphasis on safety and country of origin. We will attempt to compare our findings to the existing results to analyze the impact of the pandemic on consumer preferences. Specifically, we seek to investigate the relative importance consumers put on food safety and the change in this importance.

Over the past decades, in nations with advanced economies food safety and the corresponding consumer perception has steadily increased, resulting in a substantial trust in the food safety system, for example, in Canada [4], often due to stringent regulation and continuous inspections [5]. Nevertheless, when a food scare occurs, such as the BSE (Bovine Spongiform Encephalopathy) scandal or dioxin crises in 1999 and 2008, consumers become cautious and food safety perceptions begin to decline [6]. It is not only diseases directly impacting the meat products but also diseases emanating from or

associated with animals, such as the avian flu (H5N1) and the swine flu (H1N1), that lead to a reduction in consumption in the respective meat products and associated consumer caution [6]. With COVID-19 caused by the SARS-CoV-2 virus, probably originating from bats [7] with a suspected animal-to-human transmission in the context of a wet market [2], creating a pandemic of proportions unseen since the Spanish flu, we seek to examine what impact this pandemic may have on food safety perceptions. Our guiding research question in this regard is: How does COVID-19 impact food safety perceptions? This question translates into the following hypotheses:

**Hypotheses 1 (H1).** *Compared to the findings of Lim et al. [3], the respondents' perceived risk is higher.*

**Hypotheses 2 (H2).** *Compared to the findings of Lim et al. [3], the respondents' risk attitude is lower.*

Additionally, and perhaps most interestingly, we seek to compare the relative importance of individual beef attributes and corresponding WTP. Given the way in which the data were originally reported by Lim et al. [3], significance tests are not feasible; we therefore aim to present a descriptive side-by-side comparison of the original results from 2014 and our results from 2020. We anticipate the relative importance of food safety will increase compared to Lim et al.'s results [3], leading us to postulate that:

**Hypotheses 3 (H3).** *Compared to the findings of Lim et al. [3], the WTP for food safety attributes is higher.*

In addition to the direct comparison with the study by Lim et al. [3], we seek to examine the self-reported perceptions of the COVID-19 impact on the lives of the respondents and on beef (safety) perceptions. In the last economic downturn of the late 2000s unemployment rose substantially, and, in particular, the effects of these job losses were studied in a plethora of ways. One finding was that as a result of the recession, food insecurity was highly prevalent in the US and access to affordable and healthy food limited [8], with economic hardship and its effects on diets spreading increasingly [9]. In other words, in recessionary times unemployment is positively correlated with food insecurity [10] and with decreased food spending. Additionally, we anticipate that, in line with previous food-related crises [6] as outlined above, consumers will place a greater emphasis on food safety. We thus also formulate the following hypothesis:

**Hypotheses 4 (H4).** *The price attribute is relatively more important to respondents who are more affected by the COVID-19 pandemic.*

This paper is structured into four distinct sections. It commences by outlining the extant literature on food safety. Next, it details the employed methodology and outlines the experimental design. Thereafter, the results are presented, and the study is concluded with a discussion of these results, potential future research avenues, managerial implications, and limitations.

## 2. Literature Review: Food Safety

Over the last decades, food culture has shifted in advanced economies, often driven by both environmental and health concerns [11], to a more balanced and broad variety of food choices, such as organics. To satisfy these concerns, consumers show an increased appetite for different food attributes, such as quality [12], ascribed health benefits [13], or locality [14], and are prepared to pay a premium for these attributes [15]. One food attribute that has been somewhat less frequently studied in the context of food in advanced economies—except for food-scare crises—is that of safety. While in advanced economies the topic of food safety is generally not chiefly on consumers' minds, perhaps explained by a substantial trust in the food safety system [4], safety is of particular interest to consumers in emerging and developing economies. In China and India, for example, such trust may not exist, as evidenced by an increased WTP for dedicated food safety attributes [16,17] or given accounts where food safety is often associated with quality in emerging nations [18]. Therefore, a number of studies have given thought to the topic of food safety and even food safety knowledge in developing countries [19].

In general, it is hardly surprising that, if food safety concerns are on the consumers' minds, the majority show a marked preference, i.e., are willing to pay a premium, to reduce food safety risk, for instance, for fruit and vegetables [20]. However, when it comes to safety concerns for consumers in advanced economies, the focus tends to be slightly different, with safety aspects being primarily focused on specific considerations, such as chemicals and genetic modification, rather than food freshness, handling, and storage. In the context of food safety, meat arguably takes a special place, especially given that higher incomes globally and the rise of a middle class in emerging nations are substantial drivers of a growth in food demand [21], and, in particular, the demand for meat. In this context, there is a strong consumer preference for beef products with enhanced safety attributes [22]. Tait et al. [23] found similar results for lamb products. In developed economies such as the United States, when food safety is not a primary concern, as evidenced by a consumer belief that the US Department of Agriculture can assure food safety regarding chicken meat [24], safety is hardly on consumers' minds and even practices such as the reprocessing of meat are acceptable options [25]. This may be in stark contrast to emerging and developing nations, where skepticism regarding meat is higher, sometimes driven by limited food safety knowledge when handling meat [26].

Reflecting such different research foci in the food safety domain, we summarize several key studies in this field of research in Table 1.

**Table 1.** Summary of Relevant Food Safety Studies.

| Study | Description | Relevant Findings |
|---|---|---|
| Lim et al. [3] | WTP estimation for country of origin labeling for beef Consumer survey in the US (*n* = 1000) Choice-based experiment | Preference for domestic (US) beef and associated food safety perception Traceability and BSE-testing label increase WTP |
| Tait et al. [23] | WTP estimation for consumers of lamb meat Online survey in China, India, and the UK (n1 = 686, n2 = 695, n3 = 686) Choice-based conjoint analysis | Food safety valued highest in emerging economies Animal welfare highest in developed economies |
| Rozan et al. [27] | WTP estimation for safety of apples, potatoes, and bread Consumer sample (*n* = 120) Experimental auctions | Information about "non-safe" aspects such as GMO or heavy metal content decreased WTP |
| Amfo et al. [28] | WTP estimation for certified safe vegetables Consumer survey in Ghana (*n* = 300) Contingent valuation | Higher WTP for certified safe vegetables to avoid health-related risks Need to strengthen consumer trust in certification institutions |
| van Loo et al. [29] | WTP estimation for sustainability labels on chicken meat Belgian consumer sample (*n* = 359) Choice-based conjoint analysis | High WTP for animal welfare and free-range label Carbon footprint and organic labels not as appealing |
| Liu et al. [30] | WTP estimation for food safety of apples Chinese consumer sample (*n* = 2092) Choice-based conjoint analysis | High WTP for selected food safety attributes Importance of certifications |
| Gifford and Bernard [24] | WTP estimation for organic and natural chicken and impact estimation of label definition Lab experiment with representative US sample (*n* = 139) Experimental auctions | High trust in US food safety Negative impact of GMO on WTP |

Note: WTP = willingness to pay; BSE = bovine spongiform encephalopathy; GMO = genetically modified organism.

## 3. Materials and Methods

### 3.1. Data Collection

For this study, we employed a questionnaire survey to collect data to analyze the developed hypotheses. As our aim was to replicate parts of the study by Lim et al. [3], we also chose the United States as the research setting, collecting data through Amazon Mechanical Turk (MTurk), a commonly used crowdsourcing platform for, among other things, surveys. MTurk allows researchers to anonymously recruit study participants based on preselected criteria such as country of residence. To ensure reliable and valid results, we followed guidelines for recruiting respondents through MTurk, such as restricting survey-takers to respondents with high approval ratings [31,32]. A total of 1092 consumers responded to our survey, with 66 having to be excluded for incomplete responses and 27 for exhibiting straight-line answer patterns or failing an attention check, leaving us with a final

sample of 999 (response rate = 91.5%). Table 2 provides an overview of the socio-demographics of the sample, the original study by Lim et al. [3], and the US population. With our survey we were able to recruit consumers who more closely matched the US population than Lim et al.'s [3] in some aspects, but in terms of education our sample was skewed towards the more educated share of the US population. While it is often challenging to recruit a sample that perfectly represents the general population, we are confident our findings are representative, as MTurk exhibits few to no disadvantages compared to other methods and tends to produce reliable results [31].

**Table 2.** Sample Descriptive Statistics in %.

| Variable | Lim et al. (2014) (*n* = 1000) | This Study (2020) (*n* = 999) | US Population |
|---|---|---|---|
| **Age** | | | |
| 17–19 | 0.9 | 0.3 | 2.6 |
| 20–24 | 3.5 | 5.0 | 6.6 |
| 25–29 | 2.2 | 20.3 | 7.1 |
| 30–39 | 7.8 | 36.0 | 13.3 |
| 40–49 | 12.7 | 21.5 | 12.4 |
| 50–64 | 32.3 | 13.3 | 19.3 |
| 65+ | 40.6 | 3.5 | 16.0 |
| **Gender** | | | |
| Female | 52.5 | 41.5 | 50.8 |
| Male | 47.5 | 58.4 | 49.2 |
| Other | 0.0 | 0.1 | *n*/a |
| **Education** | | | |
| No formal education | 1.1 | 0.2 | 11.8 |
| High school | 23.1 | 6.9 | 27.5 |
| Some college | 39.4 | 14.9 | 30.7 |
| Four-year degree | 24.3 | 55.7 | 30.1 [1] |
| Graduate degree or higher | 12.1 | 21.9 | |
| Other | 0.0 | 0.4 | *n*/a |
| **Household income** | | | |
| less than USD 25,000 | 24.1 | 9.5 | 19.6 |
| USD 25,000–USD 39,999 | 23.5 | 15.2 | 13.0 |
| USD 40,000–USD 64,999 | 23.8 | 29.4 | 18.7 |
| USD 65,000–USD 79,999 | 9.6 | 17.4 | 9.5 |
| USD 80,000–USD99,999 | 7.3 | 14.3 | 10.1 |
| USD 100,000–USD 119,999 | 6.1 | 6.0 | 6.0 |
| USD 120,000 or more | 5.6 | 8.1 | 23.2 |
| **Shopping frequency** | | | |
| Never | 1.9 | 1.9 | *n*/a |
| Sometimes | 14.7 | 47.1 | *n*/a |
| Frequently | 83.4 | 51.0 | *n*/a |

Source of US population data: data.census.gov; [1] combined due to data availability.

### 3.2. Measurement Instruments and Analysis

The survey was split into three distinct sections, with the first two parts replicating the study by Lim et al. [3]. In the first section, the respondents were asked to choose among 12 choice sets that were created in XLSTAT (Version 2018.1.1.), a proprietary statistics plug-in for Microsoft Excel. The choice sets were created based on the product attributes used by Lim et al. [3] and can be found in Table 3 (examples for choice sets can also be found in Appendix A, Figure A1). In the second part, analogous to Lim et al. [3], we elicited three risk perception statements, three risk attitude statements, and country of origin perception statements, which can be seen in Table 4 in the Results section of this paper. These statements were measured on a 7-point Likert scale, with a score of 1 denoting "strongly disagree" and a score of 7 denoting "strongly agree." In addition to the choice sets and survey items

administered by Lim et al. [3], we elicited the respondents' sentiments about their food security and the perceived impact of the COVID-19 pandemic. These sentiments were also gauged by way of reflective statements measured on a 7-point Likert scale. Lastly, before asking for the socio-demographic details of the respondents, we asked the study participants to specifically report on their employment situation. These items are detailed in Table 5 in the Results section of this paper. The subsequent analyses were conducted in XLSTAT and the software solution SPSS (Statistical Package for Social Sciences, version 26). To arrive at more accurate results, we employed a 7-point Likert scale, and to compare our results, we transformed the scale used by Lim et al. [3] from a 5-point to a 7-point scale. It should be noted that the interpretability of the results was of course not influenced by this conversion.

**Table 3.** Overview of Product Attributes. Adopted from Lim et al. [3].

| Attribute | Description | Variation |
|---|---|---|
| Country of origin | Refers to country in which the cattle were raised | US<br>Canada<br>Australia |
| Production practices | Refers to the method used in production:<br>**Approved Standards** means production involved government-approved synthetic growth hormones and antibiotics<br>**Natural** means animal was raised without the use of synthetic growth hormones or antibiotics | Approved Standards<br>Natural |
| Food safety assurance | Refers to the food safety assurance offered with the steak:<br>**BSE-Tested** means that cattle are tested for BSE prior to the slaughtering process<br>**Traceable** means the product is fully traceable back to farm of origin from the point of purchase<br>**Traceable and BSE-Tested** means both processes are offered in combination | None<br>BSE-Tested<br>Traceable<br>Traceable and BSE-Tested |
| Tenderness | Refers to the softness in the steak's eating quality:<br>**Not Specified** means there are no guarantees on tenderness level of the steak<br>**Assured Tenderness** means the steak is guaranteed tender by testing the steak using a tenderness measuring instrument | Not Specified<br>Assured Tenderness |
| Price (USD/lb) | Refers to steak price in retail grocery store or butcher where the respondent typically shops | USD 5.50<br>USD 9.00<br>USD 12.50<br>USD 16.00 |

**Table 4.** Perceived Risk Statistics.

| Items | Lim et al. (2014) [1] | | This Study (2020) | |
|---|---|---|---|---|
| Risk perception | $\mu = 3.30$ | $\sigma = 1.63$ | $\mu = 3.45$ ** | $\sigma = 1.78$ |
| RP1: When eating beef, I am exposed to a great deal of risk. | $\mu = 3.40$ | $\sigma = 1.59$ | $\mu = 3.50$ | $\sigma = 1.75$ |
| RP2: I think eating beef is risky. | $\mu = 3.21$ | $\sigma = 1.67$ | $\mu = 3.42$ *** | $\sigma = 1.87$ |
| RP3: For me, eating beef is risky. | $\mu = 3.28$ | $\sigma = 1.67$ | $\mu = 3.43$ ** | $\sigma = 1.96$ |
| Risk attitude | $\mu = 4.83$ | $\sigma = 1.65$ | $\mu = 5.21$ *** | $\sigma = 1.33$ |
| RA1: I accept the risks of eating beef. | $\mu = 4.89$ | $\sigma = 1.63$ | $\mu = 5.19$ *** | $\sigma = 1.37$ |
| RA2: For me, eating beef is worth the risk. | $\mu = 4.74$ | $\sigma = 1.70$ | $\mu = 5.16$ *** | $\sigma = 1.47$ |
| RA3: I am willing to accept the risk of eating beef. | $\mu = 4.85$ | $\sigma = 1.66$ | $\mu = 5.29$ *** | $\sigma = 1.43$ |

**Table 4.** *Cont.*

| | Australia (%) | | Canada (%) | | USA (%) | |
|---|---|---|---|---|---|---|
| **Whether you have ever knowingly purchased beef produced in another country or not, what is your perception of the level of food safety of beef by country of origin?** | | | | | | |
| | Lim et al. (2014) | This study (2020) | Lim et al. (2014) | This study (2020) | Lim et al. (2014) | This study (2020) |
| very low | 6.2 | 4 | 4.8 | 4.7 | 4.3 | 5.6 |
| low | 8.1 | 8.7 | 7.1 | 7.4 | 6.0 | 6.4 |
| moderate | 23.5 | 28 | 24.9 | 22.2 | 19.7 | 17.7 |
| high | 18.9 | 30.6 | 20.9 | 34.6 | 32.2 | 27.0 |
| very high | 8.6 | 20.7 | 11.8 | 23.3 | 27.2 | 37.5 |
| no opinion | 34.7 | 7.9 | 30.5 | 7.7 | 10.8 | 5.7 |
| Mean (without "no opinion") | $\mu = 2.12$ | $\mu = 3.60$ *** | $\mu = 2.36$ | $\mu = 3.70$ *** | $\mu = 3.40$ | $\mu = 3.90$ *** |

$\mu$ = arithmetic mean; $\sigma$ = standard deviation; [1] converted to 7-point Likert scale; ** = significant at the 5% level; *** = significant at the 1% level.

**Table 5.** Self-Reported Employment Situation.

| Items | Valid % (*n* = 937) |
|---|---|
| Employment situation | |
| I am employed. | 72.0 |
| I am employed but had my hours reduced recently. | 12.8 |
| I have recently lost my job(s). | 3.8 |
| I am self-employed. | 11.3 |

Note: Only 937 of the 999 respondents reported their employment situation.

### 3.3. Experimental Design and Estimation of WTP

There are a huge number of publications available in food sciences where discrete choice modeling was used, e.g., to approximate the consumer preferences for labels [33,34], to assess the importance of local provenance of food products [35,36], or to cite specific attributes for selected meat products [37,38], to name just a few examples. One of the main advantages of discrete choice modeling is that "a stated preference discrete choice model considers a realistic buying situation, where consumers choose between one or more products from a restricted product set (evoked set)" [39]. Respondents are confronted with a limited choice of product alternatives (built with a predefined set of product attributes), therefore, the choice itself delivers binary data. The basic assumption is that respondents will choose the alternative that provides the highest utility, and a no-choice option is possible, as well, if no alternative to the choice set meets the respondents' expectations. It is then possible to approximate preferences from the buying decisions. Based on the random utility theory [40], the choice decision of consumer *j* can be expressed as

$$U_{ijs} = V_{ijs} + \varepsilon_{ijs}, \tag{1}$$

where the deterministic element $V_{ijs} = \beta_j \cdot X_{ijs}$ and $\varepsilon_{ijs}$ is the stochastic element; $X_{ijs}$ is the vector of attributes with the *i*th option of choice set *s*; $\beta_j$ is the (unknown) vector describing the preferences of the *j*th individual. In our case, we have to approximate $\beta_j$ for the Formula (2):

$$U_{ijs} = \beta_1 \cdot \text{Country of origin} + \beta_2 \cdot \text{Production practices} + \beta_3 \cdot \text{Food safety assurance} + \beta_4 \cdot \text{Tenderness} + \beta_5 \cdot \text{Price} + \varepsilon_{ijs}. \tag{2}$$

We adopted our experimental design from Lim et al. [3] to assess part-worth utilities of attributes $X_{ijs}$. Apart from the empirical design of our choice experiment, which was completely in accordance with Lim et al. [3], we used a Hierarchical Bayes (HB) estimation in order to be able to approximate individual part-worths. This approach is commonly used in consumer research as "[r]ecent advances in Bayesian estimation make the estimation of these models computationally feasible, offering advantages

in model interpretation over models based on indirect utility, and descriptive models that tend to be highly parameterized" [41]. In our case, it was necessary to approximate individual part-worths for product attributes and levels to be able to correlate the utilities with the individual perceptions of being affected by the COVID-19 pandemic.

Consumer WTP, a concept from welfare economics, "is the marginal rate of substitution of particular attributes/levels for money (price levels)" [42]. WTP can be approximated from discrete choice models when price attribute (cost) is part of the experimental design [43]. Using the additive composition rule [44] in Formula (2), the change in one specific product attribute level results in a change of the part-worths of price levels. In case of a linear utility function, $\beta_1$ is denoted as the utility per level (e.g., origin US) and $\beta_2$ as the utility per dollar. The WTP can then be interpreted as the ratio $\beta_1/\beta_2$ [42]. Therefore, if the precondition of linearity with fixed parameters holds, estimation of WTP is rather straightforward by calculating the ratio between the coefficient of the attribute (change in utility $U_i$, e.g., if country of origin shifts from US to Canada) and the cost coefficient (denoted as a linear, nonrandom coefficient) [43]. Breidert et al. [44] provided a comprehensive review of methods to approximate consumers' WTP, within which they included Jedidi and Zhang's [45] study, which presented an approach to assess "consumer-level reservation prices." Their methodological considerations help to clarify the estimation of the WTP based on discrete choice experiments. Further, confirming literature WTP approximated by choice experiments is comparable to other commonly used measures to assess WTP, like experimental auctions [46,47].

Examples of studies using discrete choice modeling approaches to assess WTP can be found in the traditional food product field [48], not only for Arctic food products [49] and ready-made food [50], but also for specific topics such as health attributes and health benefit claims [51], and food safety-enhancing technologies [22]. In this study, we will estimate WTP for all attribute levels and compare our results with those of Lim et al. [3]. Furthermore, we aim to understand, as postulated in H4, the effect of individual perceptions of being affected by the COVID-19 pandemic on the importance of the price attribute for meat products.

## 4. Results and Discussion

### 4.1. Perception of Impacts of the COVID-19 Pandemic

Regarding food security (FS), the respondents generally answered with slight disagreement about the perceived personal risks of eating beef (aggregated risk perception μ = 3.45, σ = 1.78). Compared to Lim et al. [3] (μ = 3.30, σ = 1.63), the perceived risk was significantly higher. Therefore, we were able to support H1 ($p < 0.05$), that the risk perceptions have increased significantly. Regarding consumers' risk attitudes, we observed a significant change ($p < 0.01$) compared to Lim et al. [3]. However, contrary to our hypothesis, respondents actually reported a higher willingness to accept the risks of eating beef (μ = 5.21, σ = 1.33) compared to Lim et al. [3] (μ = 4.83, σ = 1.65), thus not supporting H2. Compared to Lim et al. [3], respondents indicated a significantly higher trust in the food safety standards of all three countries (μ = 3.60 to 3.90 with 1 = "food safety is very low" to 5 = "food safety is very high"), and a much larger share of consumers now hold opinions about Canadian and Australian beef (more than nine in ten respondents now, compared to roughly two thirds before). Table 4 summarizes the perceived risk statistics and compares the findings of our sample to the findings by Lim et al. [3]. A detailed overview of the distribution of responses regarding risk statistics and all other reflective survey items can be found in the Appendix A (Table A1 in Appendix A).

In addition to the original survey items, we gauged sentiments regarding food security and the impact of the COVID-19 pandemic. We asked the respondents to report on their employment situation. Surprisingly, only 3.6% of respondents indicated that they had recently lost their jobs, well below the US average as of May 2020 [52]. This implies that people who—in socioeconomic terms—were affected the most did not take part in the study to the extent that could be expected. Table 5 summarizes these findings.

The respondents generally expressed somewhat neutral sentiments regarding food security, with a slight tendency towards perceiving food to be too expensive ($\mu = 4.76$, $\sigma = 1.24$). Regarding the impact of the pandemic, we found some individual optimism in the respondents' sentiments, as the personal optimism for the future ($\mu = 4.96$, $\sigma = 1.22$) was higher than the general optimism regarding the economy ($\mu = 4.58$, $\sigma = 1.64$). Additionally, we found that the feelings of the pandemic changing society ($\mu = 5.49$, $\sigma = 1.48$) were more pronounced than the respondents' feelings of being personally affected by the pandemic ($\mu = 5.03$, $\sigma = 1.82$). Table 6 summarizes these additional survey items.

**Table 6.** Overall Perceived Impact of COVID-19 Pandemic.

| | Total (*n* = 999) | |
|---|---|---|
| | μ | σ |
| Food security | 4.30 | 1.41 |
| FS01: I feel food is too expensive. | 4.76 | 1.24 |
| FS02: My current financial situation forced me to change my food habits. | 4.18 | 1.47 |
| FS03: I am worried about buying enough food. | 3.97 | 1.72 |
| Impact of COVID-19 pandemic | 4.96 | 0.89 |
| CI01: I feel the coronavirus pandemic has affected me personally. | 5.03 | 1.82 |
| CI02: I feel the coronavirus pandemic will change society. | 5.49 | 1.48 |
| CI03: I am optimistic regarding my financial situation. | 4.96 | 1.22 |
| CI04: I am worried about my financial future. | 4.73 | 1.41 |
| CI05: I am optimistic regarding the economy. | 4.58 | 1.64 |

μ = arithmetic mean; σ = standard deviation; data not available for Lim et al. [3].

Next, we sought to understand to what extent the employment situation affects these sentiments. As expected, the most pessimistic respondents within the sample were those who recently experienced job loss. This group of respondents seemed to be mostly affected by the COVID-19 pandemic. The differences between all four identified groups were generally significant (however, the effect size was small, with maximum $\eta^2$ amounting to 0.032 for CI04). If we go into more detail, we can observe that the group with the lowest trust in the future of the economy was self-employed. Table 7 summarizes the perceived impact of the COVID-19 pandemic by employment situation.

**Table 7.** Overall Perceived Impact of COVID-19 Pandemic and Employment Situation.

| | Employed (*n* = 675) | | Hours Reduced (*n* = 120) | | Job Loss (*n* = 36) | | Self-Employed (*n* = 106) | |
|---|---|---|---|---|---|---|---|---|
| | μ | σ | μ | σ | μ | σ | μ | σ |
| Food security *** | 4.23 | 1.42 | 4.46 | 1.31 | 5.00 | 1.34 | 4.55 | 1.38 |
| FS01: … food is too expensive. *** | 4.69 | 1.46 | 4.69 | 1.41 | 5.11 | 1.45 | 5.12 | 1.50 |
| FS02: … forced me to change my food habits. ** | 4.07 | 1.72 | 4.47 | 1.62 | 5.28 | 1.70 | 4.42 | 1.71 |
| FS03: … worried about buying enough food. | 3.94 | 1.85 | 4.23 | 1.65 | 4.61 | 1.57 | 4.11 | 1.86 |
| Impact of COVID-19 ** | 4.95 | 0.91 | 5.16 | 0.83 | 5.21 | 0.75 | 4.89 | 0.89 |
| CI01: … has affected me personally. *** | 4.92 | 1.48 | 5.38 | 1.27 | 5.86 | 0.99 | 5.11 | 1.55 |
| CI02: … will change society. ** | 5.41 | 1.25 | 5.62 | 1.09 | 5.89 | 1.01 | 5.64 | 1.17 |
| CI03: … optimistic regarding my financial situation. *** | 5.12 | 1.31 | 4.75 | 1.54 | 4.39 | 1.71 | 4.61 | 1.60 |
| CI04: … worried about my financial future. *** | 4.60 | 1.66 | 5.32 | 1.37 | 5.56 | 1.05 | 4.92 | 1.63 |
| CI05: … optimistic regarding the economy. ** | 4.68 | 1.49 | 4.73 | 1.60 | 4.36 | 1.76 | 4.17 | 1.94 |

μ = arithmetic mean; σ = standard deviation; ** = significant group differences at the 5% level; *** = significant at the 1% level.

## 4.2. Approximation of Part-Worth Utilities and Importance of Product Attributes by Choice-Based Conjoint Analysis

The next step was the choice-based conjoint analysis, where we approximated the part-worth utilities and importance of attributes as described in the Methodology section on an individual level by means of Hierarchical Bayes estimation. The relative importance of each attribute group was calculated as the difference between the highest and lowest part-worth utility in that group divided by the total

utility (i.e., the sum of all part-worth utilities for all groups). Compared to Lim et al. [3] (who estimated the part-worths by means of a Mixed Logit model), we saw that the relative importance of food safety assurances increased substantially from 24.3% to 30.1%. This increase almost exclusively coincided with a decrease in the importance of the country of origin (from 30.5% to 24.0%). The importance of the price attribute remained fairly constant (from 32.1% to 31.5%), except for the subgroup of respondents who reported having recently lost their jobs. Table 8 provides an overview of the relative importance of beef attributes.

**Table 8.** Relative Importance of Beef Attributes and Part-Worth Utilities.

| Attribute | Lim et al. (2014) [1] (*n* = 1000) | This Study (2020) Overall (*n* = 999) | This Study (2020) Hours Reduced (*n* = 120) | This Study (2020) Job Loss (*n* = 36) |
|---|---|---|---|---|
| Country of origin | 30.5% | 24.0% | 23.5% | 19.0% |
| US | 0.000 | 0.000 | 0.000 | 0.000 |
| Australia | −2.674 | −1.637 | −1.635 | −1.353 |
| Canada | −1.918 | −1.233 | −1.191 | −0.871 |
| Production practices | 0.4% | 2.4% | 2.4% | 0.4% |
| Approved Standards | 0.000 | 0.000 | 0.000 | 0.000 |
| Natural | 0.034 | 0.165 | 0.166 | 0.031 |
| Food safety assurance | 24.3% | 30.1% | 29.9% | 27.0% |
| None | 0.000 | 0.000 | 0.000 | 0.000 |
| BSE-Tested | 1.459 | 1.772 | 1.800 | 1.838 |
| Traceable | 1.526 | 1.187 | 1.208 | 1.420 |
| Traceable and BSE-Tested | 2.136 | 2.050 | 2.081 | 1.926 |
| Tenderness | 12.7% | 11.9% | 12.2% | 11.6% |
| Not Specified | 0.000 | 0.000 | 0.000 | 0.000 |
| Assured Tenderness | 1.113 | 0.812 | 0.847 | 0.827 |
| Price (USD/lb) | 32.1% | 31.5% | 32.1% | 42.0% |
| Non-random Coefficients | −0.268 | −0.207 [2] | −0.216 [2] | −0.286 [2] |

[1] Part-worth utilities taken from Lim et al. [3], relative importance estimated by part-worth utilities (not available in Lim et al. [3]). [2] Estimated out of part-worth utilities for price levels USD 5.50 to USD 16.00; aggregated results are almost perfectly linear ($R^2$ = 0.994; 0.995; 0.999).

Concerning the individual part-worth utilities of the product attributes, our analysis delivered differing results (leading to the abovementioned approximation of the importance of the product attributes). However, the general tendency of part-worth utilities was comparable to Lim et al. [3]. The origin of the beef had a negative impact if it did not come from the US; in both studies Australian beef delivered the lowest part-worth utility. Approved standards delivered only a minor plus in part-worth utility, in particular for the respondents who lost their jobs due to the COVID-19 crisis. In view of food safety issues, in both studies traceability and BSE-testing delivered the highest plus in part-worth utilities, followed by traceability in Lim et al. [3] and BSE-testing in our study. Finally, the minor plus in part-worth utility for assured tenderness was comparable in both studies.

Based on these findings, the WTP for the individual attributes was estimated. Regarding the food safety attributes, our sample exhibited a sizeable premium for safety assurances. If a traceability attribute was added, consumers would be willing to spend an additional USD 5.72 per pound of beef steak. If the beef steak was tested for BSE, the premium even exhibited amounts to USD 8.54. If the steak was both traceable and BSE-tested, consumers would pay a USD 9.88 price premium per pound. The metric size of the WTP for these attributes was comparable for traceability, but much higher in our study compared to Lim et al. [3]. Contrasting this WTP for food safety attributes, assured tenderness and natural production practices were only able to fetch a USD 3.92 premium and a USD 0.80 premium, respectively, which, again, were more or less comparable to Lim et al. [3] (USD 4.15 and USD 0.13, respectively). In view of the country of origin, a provenance other than the US was connected to a negative WTP, which was in line with Lim et al. [3]. Our results showed that food security concerns have become more important compared to Lim et al. [3], thus supporting H3, potentially due to the COVID-19 pandemic.

As demonstrated by the increased importance of the price attribute for respondents who had recently lost their jobs in Table 8, this group exhibited a lower overall WTP. This was especially evident

in the otherwise important food safety and country of origin attributes, where they exhibited a USD 7.18 premium for traceable and BSE-tested beef steak (compared to the USD 9.88 premium for the overall sample) and a negative WTP of −USD 4.73 if origin changed from US to Australia (compared to the overall sample with −USD 7.89, and also compared to those who had to reduce working hours with −USD 7.56). Table 9 summarizes the WTP for beef attributes. In accordance with Lim et al. [3], we also estimated WTP for our total sample by means of a simulation with 5000 draws to further prove the reliability of the outcome. As we can see from that, differences between the overall estimation of the total sample and the simulation were low. Also, the 95% confidence intervals were quite narrow, thus the reliability of our WTP approximation can be considered to be high. Compared to Lim et al. [3], all differences of the means were significant despite the attribute "Tenderness."

Our findings clearly showed that the COVID-19 pandemic seems to have influenced the WTP for, and evaluation of the importance of, beef attributes. This assumption is further promoted if we aggregate the results of the overall perceived COVID-19 pandemic in Table 8 with the part-worth utilities for beef attributes. We can see in Table 10 that there were significant correlations in particular between statements referring to the economic and financial situation and the expectations of respondents. The more pessimistic the respondents were, the higher they rated the importance of low prices (negative *r* in Table 10) and the more tolerant they were with respect to country of origin and food safety. In particular, all correlations between the statement "I am optimistic regarding the economy" and part-worth utilities were significant. Although Pearson's *r* was rather low for all statements, the outcome clearly showed that the individual perception of the effects of the COVID-19 pandemic significantly influenced the perception of beef attributes at least to some extent, clearly supporting H4: individuals who are more affected by the COVID-19 pandemic are becoming more price sensitive. In addition to that, they also seem to be prepared (at least to some extent) to accept the reduction of other performance attributes (like country of origin, food safety attributes).

*4.3. Discussion*

The findings are interesting in a number of ways. The topic of food safety is becoming increasingly important, compared to the findings of Lim et al. [3]. This is in line with the notion that in times of food scares, food safety perceptions begin to decline [6]. This is partly also reflected in the increase in the perceived risk the respondents reported. This leads us to conclude that consumers, at least loosely, associate the origin of the COVID-19 with meat products and are thus inclined to spend more to ensure that the purchased beef is safe for consumption. While we anticipated a decrease in consumer attitude towards accepting the risks of eating beef, we found an increased risk appetite. This increased willingness to accept the risks perhaps stems from a higher trust in the food safety standards once assurances, such as BSE-testing and traceability, are present, i.e., bought for a premium.

Another comparative finding that is interesting is the vastly increased rating of beef of Canadian and Australian origin. This might be explained by an increased level of perceived food safety and food standards, compared to the original study from 2014, perhaps due to the prevalence of such beef products and the consumers' familiarity with them.

Regarding the socioeconomic impact of the pandemic, we found that a recent job loss substantially put the price attribute into focus. In other words, the overall WTP decreased noticeably as recently unemployed respondents showed lower WTP premia for safety, tenderness, production, and origin attributes. This is in line with previous findings that in recessionary times unemployment is positively correlated with food insecurity [10] and with decreased food spending. Reduced work hours, however, had only a small negative effect on WTP. This is perhaps explained by the general optimism expressed by the respondents, who perhaps hoped that their work hours would be restored to a pre-pandemic level soon. This aforementioned optimism is best seen by the tendency to rate the personal situation, referring to both the pandemic in general and the financial outlook in particular, more favorably than that of society or the overall economy.

**Table 9.** Willingness to Pay (WTP) for Beef Attributes.

| Attributes | Lim et al. (2014): WTP | This Study (2020): WTP Overall | This Study (2020): WTP Overall [95%-Confidence Interval] [a] | This Study (2020): WTP Hours Reduced | This Study (2020): WTP Job Loss |
|---|---|---|---|---|---|
| Country of origin | | | | | |
| US (baseline) | | | | | |
| Canada | −USD 5.75 [b] | −USD 5.94 | −USD 6.20 (−6.45; −5.96) *** | −USD 5.51 | −USD 3.05 |
| Australia | −USD 7.33 [b] | −USD 7.89 | −USD 8.21 (−8.53; −7.90) *** | −USD 7.56 | −USD 4.73 |
| Production practices | | | | | |
| Approved Standards (baseline) Natural | +USD 0.13 [c] | +USD 0.80 | +USD 0.83 (0.79; 0.88) *** | +USD 0.62 | +USD 0.12 |
| Food safety assurance | | | | | |
| None (baseline) Traceable | +5.69 [c] | +USD 5.72 | +USD 5.86 (5.71; 6.01) ** | +USD 4.50 | +USD 5.29 |
| BSE-Tested | +5.44 [c] | +USD 8.54 | +USD 8.76 (8.54; 8.98) *** | +USD 6.71 | +USD 6.85 |
| Traceable and BSE-Tested | +7.96 [c] | +USD 9.88 | +USD 10.17 (9.88; 10.47) *** | +USD 7.76 | +USD 7.18 |
| Tenderness | | | | | |
| Not Specified (baseline) | | | | | |
| Assured Tenderness | +USD 4.15 [c] | +USD 3.92 | +USD 4.03 (3.91; 4.17) | +USD 3.16 | +USD 3.08 |

** = significant difference to Lim et al. [3] at the 5% level; *** = significant difference at the 1% level; [a] In accordance with Lim et al. [3]: results produced by simulation with 5000 draws; [b] WTP taken from to Lim et al. [3]; [c] not available in Lim et al. [3]; estimated by part-worth utilities (Table 6) in accordance with Section 3.3.

**Table 10.** Correlations between Part-Worth Utilities and Perceived Impact of COVID-19 Pandemic.

| Part-Worth Utility | The Pandemic Has Affected Me Personally. | | The Pandemic Will Change Society. | | I am Optimistic Regarding My Financial Situation. | | I Am Worried about My Financial Future. | | I Am Optimistic regarding the Economy. | |
|---|---|---|---|---|---|---|---|---|---|---|
| | *r* | Sig. | *r* | Sig. | *r* | Sig. | *r* | Sig. | *r* | Sig. |
| Origin US | −0.010 | 0.751 | −0.041 | 0.196 | 0.144 *** | 0.000 | 0.132 *** | 0.000 | 0.254 *** | 0.000 |
| Origin Australia | 0.023 | 0.463 | 0.037 | 0.244 | −0.102 *** | 0.001 | −0.092 *** | 0.004 | −0.206 *** | 0.000 |
| Origin Canada | −0.004 | 0.890 | 0.039 | 0.217 | −0.164 *** | 0.000 | −0.153 *** | 0.000 | −0.265 *** | 0.000 |
| Production Approved | 0.027 | 0.390 | −0.056 * | 0.078 | 0.037 | 0.244 | 0.186 *** | 0.000 | 0.129 *** | 0.000 |
| Production Natural | −0.027 | 0.390 | 0.056 * | 0.078 | −0.037 | 0.244 | −0.186 *** | 0.000 | −0.129 *** | 0.000 |
| FS None | 0.001 | 0.968 | −0.136 *** | 0.000 | 0.221 *** | 0.000 | 0.220 *** | 0.000 | 0.361 *** | 0.000 |
| FS BSE-Tested | −0.032 | 0.313 | 0.084 *** | 0.008 | −0.193 *** | 0.000 | −0.153 *** | 0.000 | −0.299 *** | 0.000 |
| FS Traceability | −0.017 | 0.597 | −0.070 ** | 0.027 | −0.114 *** | 0.000 | −0.024 | 0.449 | −0.131 *** | 0.000 |
| FS Traceability and BSE-Tested | 0.025 | 0.428 | 0.154 *** | 0.000 | −0.095 *** | 0.003 | −0.165 *** | 0.000 | −0.194 *** | 0.000 |
| Tenderness None | 0.019 | 0.554 | −0.136 *** | 0.000 | 0.136 *** | 0.000 | 0.221 *** | 0.000 | 0.255 *** | 0.000 |
| Tenderness Assured | −0.019 | 0.554 | 0.136 *** | 0.000 | −0.136 *** | 0.000 | −0.221 *** | 0.000 | −0.255 *** | 0.000 |
| Price (USD/lb) 5.5 | −0.029 | 0.357 | 0.029 | 0.367 | −0.245 *** | 0.000 | −0.129 *** | 0.000 | −0.311 *** | 0.000 |
| Price (USD/lb) 9.0 | −0.054 | 0.090 | 0.011 | 0.739 | −0.192 *** | 0.000 | −0.169 *** | 0.000 | −0.246 *** | 0.000 |
| Price (USD/lb)12.5 | 0.034 | 0.280 | −0.043 | 0.174 | 0.229 *** | 0.000 | 0.108 *** | 0.001 | 0.266 *** | 0.000 |
| Price (USD/lb) 16.0 | 0.034 | 0.278 | −0.019 | 0.545 | 0.242 *** | 0.000 | 0.149 *** | 0.000 | 0.316 *** | 0.000 |

FS = Food security; *n* = 999; Sig. = significance; * = significant at the 10% level; ** = significant at the 5% level; *** = significant at the 1% level.

### 4.4. Managerial and Policy Implications

Our findings hold several implications for food producers and retailers as well as policy-makers. For producers and retailers, the increased importance of safety attributes is a clear call to action. It might be sensible to extensively invest in traceability, testing, and labeling. Furthermore, our findings also suggest that promoting the fact that a meat product is domestic in origin might be worthwhile. For policy makers, the implications are broader and perhaps more embedded in the vast array of socioeconomic measures governments are taking in several countries. Firstly, we would urge the United States to extensively promote food safety and quality standards, perhaps by way of a tiering system that includes different safety and quality criteria. Secondly, the subsample of recently unemployed respondents being highly price-conscious likely hints at some level of food insecurity in the United States, not dissimilar to the one seen in the financial crisis that started in 2008 [8], with economic hardship and its effects on diets spreading increasingly [9]. It thus seems advisable that policy makers should ensure access to affordable and healthy food options to limit the food security effects of this pandemic-induced recession.

### 4.5. Contribution and Future Research Areas

With our study, we were able to contribute to the literature in a number of ways. For one, we were able to substantiate that, when a food-related scare occurs, consumer food safety perceptions begin to decline, in line with previous food crises [6]. Furthermore, as the pandemic and its economic consequences unfold, we were able to offer a glimpse into the implications for consumers' perceptions and WTP.

Based on our findings, we envision two main areas of future research. Firstly, as more details will undoubtedly emerge regarding the origin of the SARS-CoV-2 virus, a deeper understanding of what factors of food-related crises influence consumer perceptions can be sought; perhaps by employing multivariate techniques such as structural equation modeling. Secondly, socioeconomic crises such as this one lend themselves to interdisciplinary collaboration and research. Undoubtedly, it would be interesting to see how the looming recession further impacts food security and whether perhaps a perceived trade-off between food security and food safety will be subjected to scientific scrutiny.

### 4.6. Limitations

Finally, it should be mentioned that the present study is not without limitations. Firstly, and most importantly, the COVID-19 pandemic is still ongoing, and its true and full public health implications and economic effects will only become observable in the years to come. Secondly, while we attempted to be as thorough and exact as possible in replicating the study by Lim et al. [3], we cannot guarantee the complete absence of certain oversights or different judgment calls taking place, especially considering that not all methodological details of the original study were available. Additionally, some differences in the socio-demographic details of the two samples should be noted. Thirdly, while specifically testing for BSE was and probably still is an important signal in the beef safety domain, we can envision a more general safety attribute such as "certified safe," with a thorough explanation being able to more accurately capture the food safety perceptions of consumers. Lastly, and more generally, we would always advise researchers to conduct WTP studies in settings where real transactions take place, given the WTP overstatement bias present in hypothetical experiments [15]. Given the replicative aim of this study, however, this was not possible.

## 5. Conclusions

With this study, we set out to replicate a beef food safety study from 2014 to find out whether the COVID-19 pandemic has influenced consumer safety perceptions, given the probable genesis of the associated SARS-CoV-2 virus in the context of a wet market. In a consumer survey in the United States (*n* = 999) we were able to show that the importance of food safety attributes has increased substantially.

Additionally, with the looming recession as a strong driver, we were able to demonstrate the impact of reduced work hours and job loss on the willingness to pay (WTP) for beef. The respondents generally expressed somewhat neutral sentiments regarding food security with a slight tendency towards perceiving food to be too expensive. Regarding the impact of the pandemic, we found some individual optimism in the respondents' sentiments as the personal optimism for the future was higher than the general optimism regarding the economy. Additionally, we found that the feeling of the pandemic changing society was more pronounced than the respondents' feelings of being personally affected by the it. Lastly, perhaps not surprisingly, we found that those who recently experienced job loss seemed to be most affected by the COVID-19 pandemic. From our findings, we were able to highlight both avenues for future research and policy implications. In particular, it seems to be necessary to consider the increased importance of food safety attributes and origin of food products. Consequently, it is advisable to promote and communicate food safety and quality standards and to guarantee the access to affordable and healthy food to limit the food security effects during the pandemic.

**Author Contributions:** Conceptualization, O.M. and F.K.; methodology, O.M. and F.K.; software, O.M.; validation, O.M. and F.K.; formal analysis, O.M.; investigation, O.M. and F.K.; resources, O.M. and F.K.; data curation, F.K.; writing—original draft, O.M. and F.K.; writing—review and editing, O.M. and F.K.; visualization, O.M. and F.K.; supervision, O.M. and F.K.; project administration, O.M.; funding acquisition, O.M. All authors have read and agreed to the published version of the manuscript.

**Funding:** This research received funding from the Institute of Marketing and Innovation, University of Natural Resources and Life Sciences, Vienna, to cover publication fees.

**Acknowledgments:** We would like to thank the anonymous reviewers for their valuable comments, the respondents who participated in our survey, and also the Head of the Institute of Marketing and Innovation, Petra Riefler, for covering the publication fees.

**Conflicts of Interest:** The authors declare that there are no conflict of interest regarding the publication of this article.

## Appendix A

| | BEEF | BEEF | BEEF | |
|---|---|---|---|---|
| Country of origin | USA | Canada | Australia | |
| Production practices | Approved Standards | Natural | Natural | None of these options |
| Food-safety assurance | Traceable and BSE-tested | Traceable | Traceable and BSE-tested | |
| Tenderness | Assured Tenderness | Assured Tenderness | Assured Tenderness | |
| Price (USD /lb) | USD 16.00 | USD 9.00 | USD 5.50 | |
| Selection | ○ | ○ | ○ | ○ |

| | BEEF | BEEF | BEEF | |
|---|---|---|---|---|
| Country of origin | Australia | Australia | Canada | |
| Production practices | Natural | Approved Standards | Natural | None of these options |
| Food-safety assurance | BSE-Tested | Traceable | - | |
| Tenderness | Assured Tenderness | - | - | |
| Price (USD /lb) | USD 16.00 | USD 12.50 | USD 16.00 | |
| Selection | ○ | ○ | ○ | ○ |

| | BEEF | BEEF | BEEF | |
|---|---|---|---|---|
| Country of origin | USA | Canada | Canada | |
| Production practices | Approved Standards | Natural | Approved Standards | None of these options |
| Food-safety assurance | Traceable | Traceable and BSE-tested | BSE-Tested | |
| Tenderness | - | - | Assured Tenderness | |
| Price (USD /lb) | USD 5.50 | USD 12.50 | USD 5.50 | |
| Selection | ○ | ○ | ○ | ○ |

**Figure A1.** Examples of choice sets presented to respondents.

**Table A1.** Detailed overview of distribution of responses of reflective survey items in % (*n* = 999).

| Reflective Survey Items | 1 | 2 | 3 | 4 | 5 | 6 | 7 |
|---|---|---|---|---|---|---|---|
| **Risk perception** | | | | | | | |
| RP1: When eating beef I am exposed to a great deal of risk. | 12.9 | 24.8 | 15.4 | 12.5 | 19.2 | 11.0 | 4.1 |
| RP2: I think eating beef is risky. | 16.8 | 26.0 | 13.1 | 10.2 | 16.2 | 11.7 | 5.9 |
| RP3: For me, eating beef is risky. | 18.9 | 24.7 | 12.5 | 10.1 | 13.6 | 12.1 | 8.0 |
| **Risk attitude** | | | | | | | |
| RA1: I accept the risks of eating beef. | 2.2 | 3.4 | 5.3 | 13.8 | 27.1 | 34.2 | 13.9 |
| RA2: For me, eating beef is worth the risk. | 2.7 | 3.8 | 6.5 | 14.5 | 23.2 | 32.6 | 16.6 |
| RA3: I am willing to accept the risk of eating beef. | 2.2 | 3.8 | 5.6 | 11.2 | 24.4 | 33.4 | 19.3 |
| **Food security** | | | | | | | |
| FS01: I feel food is too expensive. | 3.1 | 6.2 | 10.3 | 14.4 | 33.1 | 24.0 | 8.8 |
| FS02: My current financial situation forced me to change my food habits. | 7.5 | 14.7 | 12.3 | 15.8 | 24.2 | 18.5 | 6.9 |
| FS03: I am worried about buying enough food. | 10.1 | 17.3 | 14.7 | 13.3 | 20.0 | 16.9 | 7.6 |
| **Impact of COVID-19 pandemic** | | | | | | | |
| CI01: I feel the coronavirus pandemic has affected me personally. | 3.0 | 6.1 | 6.6 | 8.6 | 33.1 | 30.1 | 12.4 |
| CI02: I feel the coronavirus pandemic will change society. | 0.9 | 2.7 | 2.9 | 8.0 | 30.1 | 36.0 | 19.3 |
| CI03: I am optimistic regarding my financial situation. | 2.0 | 3.8 | 9.3 | 18.1 | 26.7 | 27.6 | 12.4 |
| CI04: I am worried about my financial future. | 4.1 | 9.0 | 10.3 | 12.7 | 27.2 | 23.9 | 12.7 |
| CI05: I am optimistic regarding the economy. | 4.0 | 9.1 | 11.7 | 17.1 | 26.3 | 21.8 | 9.9 |

Note: 1 = strongly disagree, 7 = strongly agree.

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
