# Peer review of "Assessing the Impact of COVID-19 on Consumer Food Safety Perceptions—A Choice-Based Willingness to Pay Study"

_sustainability, doi:10.3390/su12187270_

Round 1
Reviewer 1 Report
- It is a good topic for this paper to analyze consumer behavior choices with COVID-19, but the authors completely use Lim et al. as the questionnaire content of this paper to analyze whether COVID-19 is appropriate? I suggest that the reason should be explained in more detail? Because The impact of BSE and COVID-19 on consumer behavior is essentially different.
- Table 5, Table 6 and Table 7 of this paper are the various effects of the COVID-19 epidemic. These parts are most closely related to the theme of this paper. It is recommended that these statements should be added to the conclusion to be more in line with the topics set by this paper .
- I suggest that references to the impact of infectious diseases on consumption should be added, such as SARS, MERS and COVID-19, which will help the completeness of this paper.
Author Response
Dear Reviewer 1,
thank you very much for your time and efforts, please address to the uploaded pdf-file for our response.
Kind regrads
Oliver Meixner & Felix Katt

Reviewer 2 Report
- Since the respondents in the research of Lim et al. (2014) are not the same as the respondents in this study it is very hard to compare the results from these two study. (Please see the table 2). However, the survey of this study provides some insight and information of beef consumption during the outbreak of COVID-19.
- Table 10. Needed to be adjusted (Colum 2)
- Please develop a respondents’ choice model for readers’ easily understandings.
- Are the food safety attributes in Lim et al. (2014) really important during the spread of COVID-19? For example, BSE is it really an issue invokes more public concerns during the spread of COVID-19 than before? The food issue that is cause more public concern should be in market channel where beef is possible to be contaminated with COVID-19 virus.
Author Response
Dear Reviewer 2,
thank you very much for your time and efforts, please address to the uploaded pdf-file for our response.
Kind regrads
Oliver Meixner & Felix Katt

Reviewer 3 Report
File enclosed

Author Response
Dear Reviewer 3,
thank you very much for your time and efforts, please address to the uploaded pdf-file for our response.
Kind regrads
Oliver Meixner & Felix Katt

Round 2
Reviewer 1 Report
The author's reply I accept in principle, the three suggestions I made during the first review, the author also did his best to make some corrections. Although some of my suggestions could not be effectively resolved, the author also explained the reasons. As a result, the paper can now be accepted for publication as is.
Author Response
Dear Reviewer 1,
Thank you very much for the time and efforts you made to review our paper. We really appreciate that.
Kind regards
Oliver Meixner & Felix Katt
Reviewer 2 Report
N.A.
Author Response
Dear Reviewer 2,
Thank you very much for the time and efforts you made to review our paper. We really appreciate that.
Kind regards
Oliver Meixner & Felix Katt
Reviewer 3 Report
- Authors have improved their manuscript consequently. Here are some minors comments.
- In 3.1. section, please correct " Generally, recruiting
respondents..." - Did authors get database of Lim et al. for the conversion of 1-5 likert scale to 1-7 likert scale ?
- I strongly recommend to authors to combine Results and Discussion sections as they entirely compared their results to Lim et al.
- In conclusion section, please put policy implications of the results of the study
Author Response
Dear Reviewer 3,
tank you very much for all the time and efforts you made to review our paper. Concerning your last remarks:
- In 3.1. section, please correct " Generally, recruiting
respondents..."
We corrected that, the sentence was replaced during the first revision, this words should have been deleted.
- Did authors get database of Lim et al. for the conversion of 1-5 likert scale to 1-7 likert scale ?
We had no access to Lim's database, but as we pointed out previously, the transformation has no impact on results and interpretation. We decided to use a 7-point scale to get more accurate results, nevertheless, a comparison with Lim et al. is still possible without influencing the outcome.
- I strongly recommend to authors to combine Results and Discussion sections as they entirely compared their results to Lim et al.
You are right, due to the nature of our contribution, we compared our results with Lim et al. during the whole chapter. We combined both chapters.
- In conclusion section, please put policy implications of the results of the study
We included the policy implications at the end of the Conclusions section: "In particular, it seems to be necessary to consider the increased importance of food safety attributes and origin of food products. Consequently, it is advisable to promote and communicate food safety and quality standards and to guarantee the access to affordable and healthy food to limit the food security effects during the pandemic."